# AdaMoLE: Fine-Tuning Large Language Models with Adaptive Mixture of Low-Rank Adaptation Experts

**Zefang Liu**
School of Computational Science and Engineering
Georgia Institute of Technology
Atlanta, GA 30332, USA
`liuzefang@gatech.edu`

**Jiahua Luo**[*]
Department of Computer and Information Science
University of Macau
Taipa, Macau, China
`luojiahuaguet@163.com`

## Abstract

We introduce AdaMoLE, a novel method for fine-tuning large language models (LLMs) through an Adaptive Mixture of Low-Rank Adaptation (LoRA) Experts. Moving beyond conventional methods that employ a static top-k strategy for activating experts, AdaMoLE dynamically adjusts the activation threshold using a dedicated threshold network, adaptively responding to the varying complexities of different tasks. By replacing a single LoRA in a layer with multiple LoRA experts and integrating a gating function with the threshold mechanism, AdaMoLE effectively selects and activates the most appropriate experts based on the input context. Our extensive evaluations across a variety of commonsense reasoning and natural language processing tasks show that AdaMoLE exceeds baseline performance. This enhancement highlights the advantages of AdaMoLE's adaptive selection of LoRA experts, improving model effectiveness without a corresponding increase in the expert count. The experimental validation not only confirms AdaMoLE as a robust approach for enhancing LLMs but also suggests valuable directions for future research in adaptive expert selection mechanisms, potentially broadening the scope for optimizing model performance across diverse language processing tasks.

## 1 Introduction

The evolution of large language models (LLMs) has been a cornerstone in the advancement of natural language processing (NLP), enabling an unprecedented depth of understanding and generation of human language. Fine-tuning these sophisticated models is essential for tailoring their capabilities to specific tasks, thereby enhancing their applicability and performance across a spectrum of NLP challenges. Despite significant progress, conventional fine-tuning methods often lack the dynamism to adapt to the diverse and complex nature of various language tasks, highlighting the need for more flexible and adaptable fine-tuning strategies.

Amidst the quest for efficiency in fine-tuning LLMs, the concept of parameter efficiency has gathered attention, particularly due to the vast size and intricate architecture of modern models. Parameter-efficient fine-tuning (PEFT) approaches (Liu et al., 2022) aim to adapt LLMs to specialized tasks by fine-tuning a small subset of model parameters, significantly reducing computational and storage costs while mitigating the risk of catastrophic forgetting.

---

[*]Corresponding author

Among these approaches, Low-Rank Adaptation (LoRA) (Hu et al., 2021) is notable for its ability to introduce adaptability without altering the original model's weights. LoRA applies low-rank decomposition to represent weight updates through smaller matrices, allowing the model to adapt to new data while the core weight matrix remains unchanged, thus embodying a targeted and efficient method for model refinement.

Building on this foundation, recent advancements have combined LoRA with the Mixture of Experts (MoE) (Shazeer et al., 2017) framework to further enhance the model's adaptability and performance. LoRA's integration allows for precise modification of weights through low-rank matrices, while MoE leverages a set of expert networks, each specializing in different tasks or aspects of the data. The synergy between LoRA's targeted weight adaptation and MoE's expert-driven approach offers a dynamic avenue for model enhancement. However, the prevalent static top-k expert selection in MoE does not fully leverage the potential for task-specific adaptability, prompting the need for more dynamic selection mechanisms that can respond to the varying complexities and subtleties of different tasks and contents.

Addressing this gap, we present AdaMoLE[1], a novel method that synergizes LoRA with an adaptive MoE, incorporating a dynamic threshold network for expert activation, which is illustrated in Figure 1. This innovation allows AdaMoLE to fine-tune its activation of experts based on the context of the input, providing a more refined and context-aware approach to model adaptation.

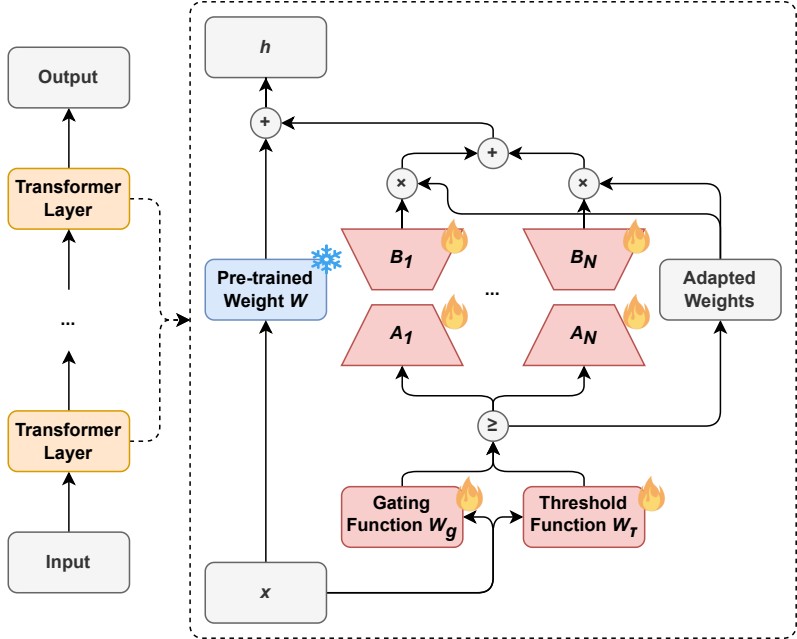

Figure 1: Illustration of Adaptive Mixture of Low-Rank Adaptation Experts (AdaMoLE). AdaMoLE employs a gating function alongside a threshold function to determine the activation of experts. In the training phase, pre-trained weights are frozen while the LoRA experts and two functions are updated.

Our main contributions are as follows:

1. AdaMoLE represents an advanced integration of LoRA and an adaptive MoE framework, featuring a dynamic threshold network that facilitates context-sensitive expert activation, transcending the limitations of static top-k strategies.
2. Through comprehensive evaluations across various tasks, AdaMoLE showcases superior adaptability and performance, highlighting the effectiveness of dynamic expert selection and setting a new baseline in the fine-tuning of LLMs.

---

[1]GitHub: https://github.com/zefang-liu/AdaMoLE

3. Threshold sensitivity and expert activation analyses of AdaMoLE provide crucial insights into the model's operational dynamics, confirming that its adaptive threshold mechanism plays a pivotal role in balancing computational efficiency with expert engagement across diverse tasks.

By introducing AdaMoLE, we aim to not only refine the fine-tuning process for LLMs but also encourage further research in developing models that are inherently flexible and attuned to the specificities of diverse application domains. This work reflects our commitment to enhancing the capabilities of LLMs, suggesting a promising direction for increased personalization and efficiency in NLP.

## 2 Related Work

The intersection of Mixture of Experts (MoE) and Low-Rank Adaptation (LoRA) in enhancing large language models (LLMs) has been a prominent area of research. Here, we explore recent works related to our AdaMoLE model.

### 2.1 Integration of MoE and LoRA

The integration of Mixture of Experts (MoE) and Low-Rank Adaptation (LoRA) is a notable trend in recent advancements aimed at enhancing the performance of LLMs. Zadouri et al. (2023) introduce a novel, parameter-efficient MoE framework, Mixture of Vectors (MoV) and Mixture of LoRA (MoLORA), designed for constrained environments, achieving performance comparable to full fine-tuning with significantly fewer parameter updates. Huang et al. (2023) investigate the composability of LoRA in LoraHub, a framework designed for cross-task generalization through the dynamic assembly of LoRA modules, aiming to achieve adaptable performance on unseen tasks. Wu et al. (2023) present Mixture of LoRA Experts (MoLE), a method that combines multiple LoRA modules within an MoE framework to improve task performance through hierarchical control and branch selection. Dou et al. (2023) introduce LoRAMoE, which integrates several LoRA adapters within an MoE-style plugin to alleviate world knowledge forgetting in LLMs during supervised fine-tuning. While these studies contribute valuable insights into the fusion of MoE and LoRA, our AdaMoLE framework stands out by implementing a dynamic thresholding mechanism, which offers a more context-responsive and flexible strategy for expert activation, optimizing the fine-tuning process across various tasks.

### 2.2 Adaptive MoE Approaches

The adaptability within Mixture of Experts (MoE) architectures is a key focus of recent research, aimed at enhancing the flexibility and efficiency of large model training and inference. Li et al. (2023) introduce an innovative adaptive gating mechanism for MoE-based language models, which adjusts the number of experts processing each token based on its linguistic complexity, a step toward optimizing computational costs. Chen et al. (2023) present AdaMV-MoE, a framework that dynamically modulates the number of active experts according to the complexity of the task at hand, underscoring the importance of adaptiveness in multi-task learning environments. Furthering this theme, Gou et al. (2023) develop MoCLE, an MoE architecture that activates parameters based on instruction clusters, enhancing the model's adaptability to diverse tasks. Moreover, Gao et al. (2024) introduce a novel MoE-LoRA method with layer-wise expert allocation, MoLA, demonstrating that more LoRA experts in higher layers can significantly enhance the performance of transformer-based models. While these advancements mark significant progress in adaptive MoE models, AdaMoLE distinguishes itself by employing a dynamic threshold network to fine-tune expert selection in real-time, ensuring that each input is processed by the most appropriate experts. Unlike the adaptive gating in Li et al. (2023), which relies on a fixed threshold to determine expert involvement, AdaMoLE's thresholding is contextually driven, offering a more granular and responsive approach to expert activation, tailored to the specific requirements of each input.

# 3 Methodology

This section delves into the core components of our proposed AdaMoLE framework, beginning with an overview of the foundational technologies it builds upon and then introducing the main design behind the AdaMoLE.

## 3.1 Preliminaries

**Low-Rank Adaptation (LoRA):** LoRA (Hu et al., 2021) is a distinguished method for enhancing parameter efficiency in the fine-tuning of large language models (LLMs). It innovatively employs low-rank matrix factorization to modify the weight matrices within a pre-trained model. For a given linear layer with weights $W_0 \in \mathbb{R}^{d \times k}$, LoRA introduces two lower-rank matrices, $A \in \mathbb{R}^{r \times k}$ and $B \in \mathbb{R}^{d \times r}$, where $r \ll \min(d, k)$ denotes the rank. The modification does not alter $W_0$ directly but adds a rank-constrained update $BA$, where the product $BA$ is the low-rank approximation that modifies the output:

$$h = W_0 x + \Delta W x = W_0 x + BAx. \tag{1}$$

During training, $A$ and $B$ are adjusted while $W_0$ remains constant, thus allowing for efficient fine-tuning. The initialization of $A$ typically follows a random Gaussian distribution, and $B$ starts from zero, ensuring that the initial state mirrors the pre-trained model's output. The $\Delta W x$ is then scaled by $\alpha / r$, where $\alpha$ is a constant.

**Mixture of Experts (MoE):** The MoE (Shazeer et al., 2017; Fedus et al., 2022a; Zoph et al., 2022; Lepikhin et al., 2020; Fedus et al., 2022b) framework scales model complexity and capacity by integrating multiple expert sub-networks, each potentially specializing in different data segments or tasks. Within a MoE layer, $N$ independent experts $\{E_i\}_{i=1}^N$ are coordinated by a router, which employs a trainable matrix $W_g$ to distribute the input vector $x$ among these experts. This router generates a distribution of weights across the experts for each input, using a softmax function for normalization:

$$p_i = \text{Softmax}(W_g x)_i. \tag{2}$$

The resultant output from the MoE layer is a weighted sum of the outputs from the top $K$ experts:

$$y = \sum_{i=1}^N \frac{\text{TopK}(p_i)}{\sum_{i'=1}^N \text{TopK}(p_{i'})} \cdot E_i(x), \tag{3}$$

where the TopK function identifies and retains the highest $K$ weights, setting the rest to zero. The weights retained by the TopK function are subsequently normalized to ensure their sum is one. Moreover, a load balancing loss following Switch Transformers (Fedus et al., 2022b) is applied to encourage an even distribution of input across experts, promoting diverse utilization of the available expertise.

## 3.2 AdaMoLE

Given the varying complexity of contexts and tasks, it is intuitive that some would require more expert involvement than others. To accommodate this, we propose an adaptive method that adjusts the number of engaged experts in each MoE layer dynamically, rather than relying on a fixed top-k selection.

Traditionally, MoE layers select experts based on the highest weights, typically choosing top-1 or top-2 experts as determined by $\text{TopK}(p_i)$. A flexible approach can be implemented by defining a threshold $\tau$ for the weights, selecting expert $i$ if $p_i \geq \tau$. The choice of $\tau$ is critical; a very high $\tau$ could result in no experts being selected if all weights $p_i$ fall below the threshold. To mitigate this, we set $\tau = 1/N$ as a lower bound of expert weights, ensuring at least one expert is selected. This is because if all $p_i < \tau$, then $\sum_{i=1}^N p_i < N\tau = 1$, contradicting the fact that the sum of expert weights $\sum_{i=1}^N p_i$ must equal 1. Consequently, the output from the MoE layer with thresholding is calculated by

$$y = \sum_{i=1}^N \frac{\mathbb{1}(p_i \geq \tau) \cdot p_i}{\sum_{i'=1}^N \mathbb{1}(p_{i'} \geq \tau) \cdot p_{i'}} \cdot E_i(x), \tag{4}$$

where $\mathbb{1}(\text{condition})$ equals 1 if the condition holds true and 0 otherwise. This threshold-based selection ensures that the number of experts is dynamically adjusted to suit the task's demands, enhancing the MoE layer's adaptability and effectiveness.

However, the fixed threshold approach lacks the flexibility required for the dynamic nature of various contexts. To address this, we introduce an adaptive MoE, where the static threshold is substituted with a threshold function. Specifically, we employ a single linear layer followed by a sigmoid function to determine the threshold:

$$\tau = \tau_{\max}\sigma(W_\tau x + b_\tau), \tag{5}$$

where $\tau_{\max}$ is the maximum threshold and can be set as $1/N$. This adaptive threshold allows for context-aware determination of the number of experts to engage. However, while introducing this adaptive threshold, it is crucial to ensure the learnability of the threshold function parameters during backpropagation. Therefore, we refine the weighted output from the experts as follows:

$$y = \sum_{i=1}^{N} \frac{\mathbb{1}(p_i \geq \tau)(p_i - \tau)}{\sum_{i'=1}^{N} \mathbb{1}(p_{i'} \geq \tau)(p_{i'} - \tau)} \cdot E_i(x), \tag{6}$$

where $p_i$ in the previous formula was replaced by $p_i - \tau$.

Incorporating the LoRA module, we derive the Adaptive Mixture of LoRA Experts (AdaMoLE):

$$h = W_0 x + \sum_{i=1}^{N} \frac{\mathbb{1}(p_i \geq \tau)(p_i - \tau)}{\sum_{i'=1}^{N} \mathbb{1}(p_{i'} \geq \tau)(p_{i'} - \tau)} \cdot B_i A_i x, \tag{7}$$

where each pair $(A_i, B_i)$ corresponds to a different LoRA expert. AdaMoLE's adaptive thresholding mechanism offers significant advantages over previous methods. By dynamically adjusting the number of engaged experts based on the input context, AdaMoLE ensures that the model's capacity is utilized more efficiently and effectively, enhancing its ability to tackle a wide array of tasks and contents with varying complexity.

## 4 Experiments

In this section, we present the comprehensive evaluation of AdaMoLE, detailing the baseline models, benchmark datasets, experimental setup, and performance outcomes.

### 4.1 Baseline Models

To benchmark AdaMoLE's effectiveness, we compare its performance with several baseline models: Low-Rank Adaptation (LoRA) (Hu et al., 2021), Sparse Mixture of Low Rank Adaptation (SiRA) (Zhu et al., 2023), three configurations of the Mixture of LoRA Experts (MoLE) (Wu et al., 2023; Gao et al., 2024). LoRA, a method that applies low-rank matrix updates for model fine-tuning, serves as the initial baseline. SiRA applies the Sparse Mixture of Expert (SMoE) for increasing the LoRA performance. The first two MoLE variants activate the top-2 and top-3 experts respectively among $N$ experts based on their weights, embodying the original MoLE approach. The third MoLE variant employs a hard thresholding strategy, selecting experts whose weights exceed the fixed threshold $\tau$ of $1/N$, introducing a basic level of adaptiveness into the expert selection process. These baseline models, with their distinct mechanisms for expert activation, provide a comprehensive backdrop to evaluate AdaMoLE's dynamic thresholding approach.

### 4.2 Benchmark Datasets

In this study, we conduct a thorough evaluation of AdaMoLE using a meticulously selected array of benchmark datasets, concentrating on two pivotal areas: commonsense reasoning and natural language processing (NLP) tasks, to thoroughly probe and ascertain the model's cognitive prowess and linguistic agility. For commonsense reasoning, we employ a diverse suite of datasets, including CommonsenseQA (Talmor et al., 2018), which challenges

the model's grasp of everyday knowledge; Cosmos QA (Huang et al., 2019), which tests inferential reasoning through narrative comprehension; and SocialIQA (Sap et al., 2019), PhysicalIQA (Bisk et al., 2020), and ScienceQA (Lu et al., 2022), which respectively examine the model's understanding of social interactions, physical phenomena, and scientific principles, each demanding a deep-seated comprehension of factual content and deductive logic.

Our investigation also extends to encompass NLP tasks from the SuperGLUE benchmark (Wang et al., 2019), renowned for its stringent standards, containing BoolQ (Clark et al., 2019), which assesses the model's ability to evaluate the veracity of statements; CB (De Marneffe et al., 2019), which probes into textual entailment and contradiction; COPA (Roemmele et al., 2011), which tests causal reasoning; RTE (Dagan et al., 2005; Haim et al., 2006; Giampiccolo et al., 2007; Bentivogli et al., 2009), which evaluates the model's ability to recognize textual entailment; and WiC (Pilehvar & Camacho-Collados, 2018), which examines the model's proficiency in interpreting word meanings across varying contexts. We reformat all benchmark datasets into a multiple-choice format (Gao et al., 2021), structuring each item with a distinct question and a set of choices, to ensure uniformity and enhance the evaluation process across different tasks.

This extensive and multifaceted evaluation is designed to provide a comprehensive assessment of AdaMoLE's fine-tuning effectiveness, illustrating its capacity to adapt and excel across a broad spectrum of commonsense reasoning and NLP tasks. For benchmark evaluations, we employ accuracy as the primary metric, selecting the choices with the highest probability from the causal language model to determine the model's performance across various tasks.

### 4.3 Experiment Settings

In the experimental setup, we employ Llama-2-7B (Touvron et al., 2023) as the foundation model. For the LoRA (Hu et al., 2021) baseline, each LoRA module is configured with a rank of 32, an alpha of 16, and a dropout rate of 0.05. In contrast, for the Mixture of Experts (MoE) models, which include both MoLE and AdaMoLE, we utilize 8 LoRA experts, with each expert having a rank of 4, ensuring that the total LoRA rank remains consistent at 32 across all models to maintain parameter parity, besides parameters in the gating and threshold functions. The adaptation is specifically targeted at four weight matrices in the self-attention module ($W_q$, $W_k$, $W_v$, $W_o$) of Llama-2, focusing our fine-tuning efforts on key components of the model's architecture. Input sequences are truncated to a maximum length of 256 tokens by following Sanh et al. (2021) and Gao et al. (2024). Training is conducted with the AdamW (Loshchilov & Hutter, 2017) optimizer, a total batch size of 16, and a learning rate of 1e-4, employing a constant learning rate scheduler with an initial warm-up phase of 200 steps to stabilize the training process. The number of training epochs is tuned between 1 to 20 by the validation accuracy on the specific dataset, allowing for flexible adjustment to optimize performance. An auxiliary loss coefficient of 1e-3 is applied to the load balancing loss (Fedus et al., 2022b) to ensure effective distribution of computation across experts. We utilize models and frameworks provided by Hugging Face's Transformers and PEFT libraries (Wolf et al., 2019; Mangrulkar et al., 2022). All experiments are conducted on a single NVIDIA H100 GPU.

### 4.4 Experiment Results

The performance of AdaMoLE, as shown in Tables 1 and 2, demonstrates its advantage over traditional baselines, with notable accuracy improvements across both commonsense reasoning and NLP tasks. These tables highlight AdaMoLE's capacity to handle a variety of datasets, showcasing its effective logical reasoning and language comprehension.

In the realm of commonsense reasoning, as detailed in Table 1, AdaMoLE demonstrates a consistent lead over the LoRA baseline and MoLE variations on multiple benchmarks, including CommonsenseQA, SocialIQA, and ScienceQA, showcasing its aptitude for complex query processing and deep reasoning. Furthermore, AdaMoLE asserts itself as a close contender in Cosmos QA and PhysicalIQA, trailing just behind the top performer, which

| Model | Gating | CommonsenseQA | Cosmos QA | SocialIQA | PhysicalIQA | ScienceQA |
|---|---|---|---|---|---|---|
| LoRA | - | 76.25% | 82.71% | 76.25% | **83.08%** | 88.67% |
| SiRA | top-2 | 73.55% | 82.18% | 76.77% | 82.59% | 86.29% |
| MoLE | top-2 | 77.15% | 83.48% | 76.92% | 82.81% | 88.71% |
| MoLE | top-3 | 77.07% | 83.48% | 77.02% | 82.10% | 90.02% |
| MoLE | $1/N$ | 75.35% | **84.69%** | 76.51% | 82.43% | 90.62% |
| AdaMoLE | 0-1/$N$ | **78.71%** | 84.25% | **77.28%** | 82.92% | **91.00%** |

Table 1: Experiment results on commonsense reasoning benchmarks, where the MoLE gating is the threshold number for expert activation, the AdaMoLE gating is the threshold range, $N$ is the number of experts in one MoE module, and the evaluation metric is accuracy.

| Model | Gating | BoolQ | CB | COPA | RTE | WiC |
|---|---|---|---|---|---|---|
| LoRA | - | 80.28% | 71.43% | 93.00% | 73.65% | 62.07% |
| SiRA | top-2 | 80.28% | **73.21%** | 92.00% | 77.26% | 57.21% |
| MoLE | top-2 | 84.37% | 69.64% | 92.00% | 86.28% | 70.06% |
| MoLE | top-3 | 86.12% | 71.43% | 92.00% | 84.84% | 62.85% |
| MoLE | $1/N$ | 84.98% | 69.64% | 93.00% | 86.28% | 69.12% |
| AdaMoLE | 0-1/$N$ | **86.27%** | **73.21%** | **94.00%** | **87.73%** | **70.22%** |

Table 2: Experiment results on NLP benchmarks from SuperGLUE, where the MoLE gating is the threshold number for expert activation, the AdaMoLE gating is the threshold range, $N$ is the number of experts in one MoE module, and the evaluation metric is accuracy.

attests to its robust performance even in domains where it does not take the top position. This performance underscores AdaMoLE's sophisticated understanding of commonsense reasoning and its strategic allocation of expertise to tackle the intricate challenges presented by these tasks.

Moving to the NLP tasks as shown in Table 2, AdaMoLE's strong performance is evident. It shows impressive accuracies in BoolQ and COPA, emphasizing its effectiveness in interpreting the veracity of statements and analyzing causal relationships. Notably, AdaMoLE shines in CB and RTE, where it explicitly outperforms the baselines, showcasing its ability to discern textual entailment with a high degree of accuracy. Similarly, in WiC, AdaMoLE demonstrates its proficiency in understanding word meanings in various contexts. These outcomes serve as a testament to AdaMoLE's strategic application of dynamic thresholding, which adeptly engages the most appropriate LoRA experts in response to the demands of each unique task, illustrating its refined adaptability in language comprehension.

Overall, AdaMoLE's remarkable performance across these diverse tasks illustrates its broad applicability and potential to enhance fine-tuning methods for large language models. This progress not only highlights AdaMoLE's adaptability but also underscores the possibility for further advancements in language model fine-tuning, particularly through the strategic use of dynamic thresholding to meet the complex demands of advanced language processing tasks.

## 5 Model Analyses

In this section, we conduct a detailed examination of AdaMoLE's threshold sensitivity, expert activation behavior, and hyperparameter settings, offering insights into how these elements influence the model's overall performance.

### 5.1 Threshold Sensitivity Analysis

The threshold sensitivity analysis investigates how varying the threshold range in AdaMoLE influences its performance on commonsense reasoning and natural language processing (NLP) benchmarks. The results in Table 3 demonstrate that AdaMoLE achieves optimal

performance on CommonsenseQA and COPA when the threshold is set within $[0, 1/(2N)]$, where $N$ is the number of experts in each MoE module. It is noteworthy that setting the threshold too high such as $[0, 1]$ leads to a significant drop in performance, since adaptive thresholds can potentially exceed most expert weights, risking no expert activation and reducing the model to its base performance without LoRA enhancements. Conversely, a lower threshold range like $[0, 1/(2N)]$ allows for more expert activations, potentially improving performance by leveraging a broader range of expertise at the cost of increased computational demand during inference. This intricate interaction between threshold selection and expert activation highlights the critical need for setting the proper threshold parameter, aiming to find the suitable balance that maximizes model performance.

| Model | Gating | CommonsenseQA | COPA |
|---|---|---|---|
| LoRA | - | 76.25% | 93.00% |
| MoLE | top-1 | 74.04% | 90.00% |
| MoLE | top-2 | 77.15% | 92.00% |
| MoLE | top-3 | 77.07% | 92.00% |
| MoLE | $1/N$ | 75.35% | 93.00% |
| AdaMoLE | $0\text{-}1/(2N)$ | **78.95%** | **96.00%** |
| AdaMoLE | $0\text{-}1/N$ | 78.71% | 94.00% |
| AdaMoLE | $0\text{-}3/(2N)$ | 77.15% | 92.00% |
| AdaMoLE | $0\text{-}2/N$ | 74.45% | 89.00% |
| AdaMoLE | $0\text{-}1$ | 31.94% | 57.00% |

Table 3: Evaluation of AdaMoLE's performance with various threshold settings, where $N$ is the number of experts in one MoE module.

Additional insights into AdaMoLE's expert utilization can be gleaned from the analysis of activated expert counts as shown in Table 4. Notably, the threshold set at $[0, 1/N]$ activates more experts than the top-2 gating but achieves better performance metrics, highlighting its efficiency in leveraging expert contributions. In contrast, the $[0, 3/(2N)]$ threshold activates fewer experts than the MoLE top-2 configuration yet manages to deliver comparable performance. This pattern suggests that AdaMoLE can maintain competitive performance even with fewer experts activated, demonstrating the model's effective balance between computational efficiency and expert deployment. While the current study focuses on AdaMoLE's adaptive thresholding mechanism, future work could provide a more detailed comparison of its performance and computational cost with other approaches such as Sparse Mixture of Low Rank Adaptation (SiRA) (Zhu et al., 2023) and Chain of LoRA (COLA) (Xia et al., 2024).

| Model | Gating | CommonsenseQA | COPA |
|---|---|---|---|
| AdaMoLE | $0\text{-}1/(2N)$ | 6.59 | 6.86 |
| AdaMoLE | $0\text{-}1/N$ | 3.46 | 4.56 |
| AdaMoLE | $0\text{-}3/(2N)$ | 1.26 | 1.71 |
| AdaMoLE | $0\text{-}2/N$ | 0.33 | 0.33 |

Table 4: Averaged numbers of activated experts from MoE modules in AdaMoLE with various threshold settings, where $N$ is the number of experts in one MoE module.

## 5.2 Expert Activation Analysis

The analysis of expert activation within AdaMoLE reveals insightful trends in how the model's LoRA experts are engaged across different tasks. For both CommonsenseQA and COPA, depicted in Figure 2, there is a noticeable trend of higher expert activation in the lower layers of the Llama-2-7B model. This suggests that the initial layers are crucial for processing foundational language features, which may be more complex and varied, necessitating a broader range of expert knowledge. As the input passes through successive layers, a reduced number of experts is engaged, indicating a refinement in processing where

fewer specialized contributions are required. The distinct activation patterns across the self-attention module's weight matrices further reflect the model's strategic allocation of expertise to handle the task-specific complexities encountered at each layer.

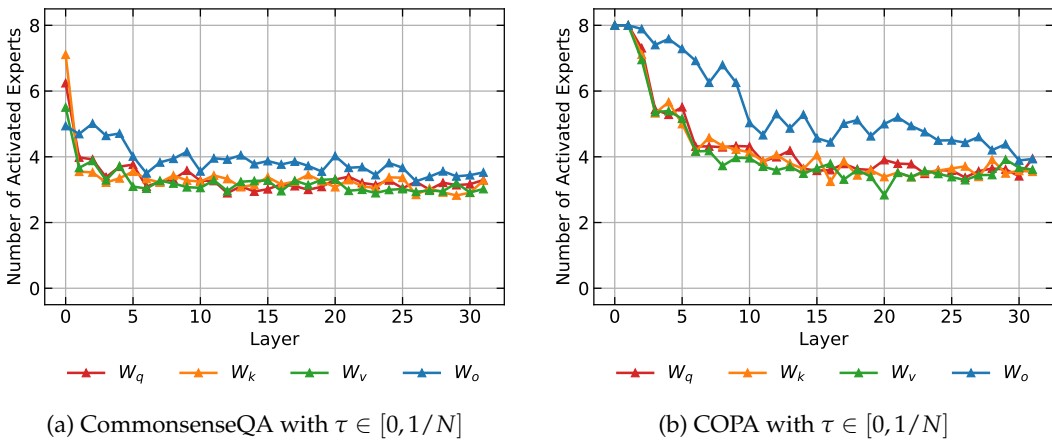

(a) CommonsenseQA with $\tau \in [0, 1/N]$        (b) COPA with $\tau \in [0, 1/N]$

Figure 2: Numbers of activated LoRA experts in AdaMoLE for four weight matrices in the self-attention module of each layer, where $\tau$ is the threshold for expert activation and $N$ is the number of experts in one MoE module.

Building on the initial findings, we further explore how the upper bounds of threshold settings affect expert activation in AdaMoLE. Figure 3 shows that as the upper bound for the threshold $\tau_{max}$ increases, the model tends to activate fewer experts, suggesting a more selective and targeted utilization of expertise. This trend is evident in both CommonsenseQA and COPA tasks and illustrates a key aspect of AdaMoLE's design, where it is able to dynamically adjust expert involvement not only enhances model performance but also optimizes computational efficiency. Striking a balance between these two factors is critical; too broad a threshold may underutilize available expertise, while too conservative a threshold can lead to computational waste. Thus, fine-tuning the threshold bounds in AdaMoLE becomes a pivotal strategy in future research, ensuring it leverages the right amount of expertise to effectively process complex tasks without incurring unnecessary computational costs.

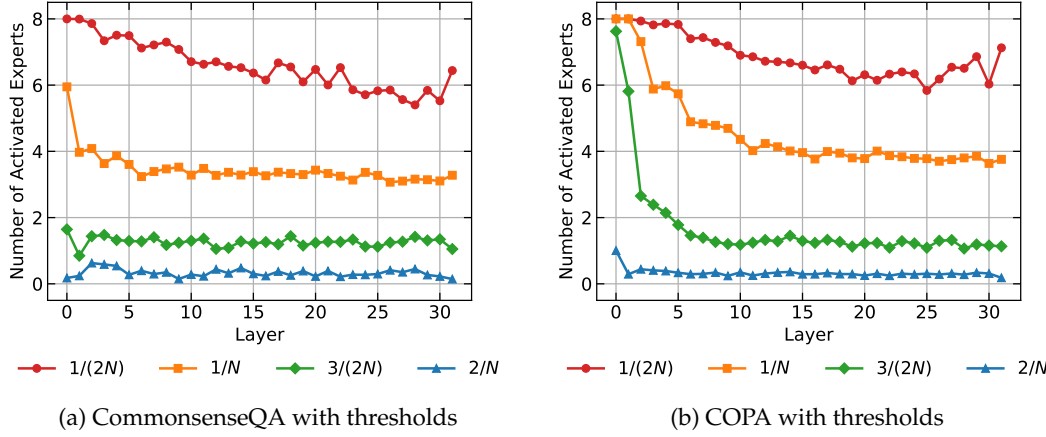

(a) CommonsenseQA with thresholds        (b) COPA with thresholds

Figure 3: Averaged numbers of activated LoRA experts in AdaMoLE for each layer with different upper bounds $\tau_{max}$, where the expert activation threshold $\tau \in [0, \tau_{max}]$ and $N$ is the number of experts in one MoE module.

### 5.3 Hyperparameter Setting Analysis

In our pursuit to optimize AdaMoLE's configuration, we carry out a systematic exploration of various hyperparameter settings on the combination of the number of experts and the LoRA rank ($N \times r$). The outcomes with the CommonsenseQA benchmark, as delineated in Table 5, demonstrate that different configurations impact AdaMoLE's performance. Our experiments reveal that AdaMoLE consistently achieves its best performance across various settings, surpassing both standard LoRA and MoLE under similar conditions. This analysis not only helps in identifying the most effective hyperparameter combinations but also underscores the adaptability of AdaMoLE to maintain superior performance across a range of expert and rank configurations, validating its robustness and effectiveness in fine-tuning LLMs.

| Model | Gating | $4 \times 4$ | $4 \times 8$ | $8 \times 4$ | $8 \times 8$ | $16 \times 4$ |
|---|---|---|---|---|---|---|
| LoRA | - | 75.43% | 76.25% | 76.25% | 76.74% | 76.74% |
| MoLE | top-2 | 73.55% | 74.45% | 77.15% | 76.82% | 76.66% |
| MoLE | $1/N$ | 74.77% | 74.28% | 75.35% | 68.88% | 75.18% |
| AdaMoLE | $0\text{-}1/N$ | **78.38%** | **77.89%** | **78.71%** | **78.13%** | **78.13%** |

Table 5: Experiment results with different hyperparameter settings of the number of experts and the LoRA rank ($N \times r$) on CommonsenseQA, where the evaluation metric is accuracy.

## 6 Conclusion

In conclusion, this study presents AdaMoLE, an innovative method that combines Low-Rank Adaptation (LoRA) with an adaptive Mixture of Experts (MoE) framework, marking a significant progress in the fine-tuning of large language models (LLMs). Through comprehensive testing on a variety of commonsense reasoning and natural language processing (NLP) tasks, AdaMoLE has proven to surpass conventional baselines, including standard LoRA and Mixture of LoRA Experts (MoLE) configurations. AdaMoLE's effectiveness is primarily attributed to its dynamic thresholding mechanism, which precisely adjusts the activation of LoRA experts based on the context of each input, ensuring the most effective use of expert knowledge. This mechanism allows AdaMoLE to achieve improvements in performance across various tasks, demonstrating its capability as an effective instrument for enhancing LLMs. Insights from our threshold sensitivity, expert activation, and hyperparameter setting analyses further highlight AdaMoLE's strategic adaptability and expert utilization, reinforcing its strong performance. Looking ahead, AdaMoLE encourages further exploration into adaptable fine-tuning methods, suggesting opportunities for continued advancements in the domain of NLP.

## Limitations

While AdaMoLE demonstrates improvements in fine-tuning LLMs by dynamically selecting LoRA experts, it has several limitations. First, the computational overhead introduced by the adaptive thresholding mechanism may still be non-trivial, particularly for huge models or when deploying in resource-constrained environments. Second, the evaluation is limited to specific benchmarks for commonsense reasoning and NLP tasks, which may not fully capture the model's performance across all potential applications and specific domains. Additionally, the current implementation of AdaMoLE does not account for the potential interactions between experts, which might lead to suboptimal expert activations in some contexts. Further research is needed to explore more sophisticated mechanisms for expert selection and activation that could mitigate these issues and improve computational efficiency.

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

# A Foundation Model Analysis

To thoroughly assess the adaptability and performance of AdaMoLE across different foundational architectures, we conduct additional experiments using Gemma-7B (Team et al., 2024) and Llama-2-13B (Touvron et al., 2023) models. The findings, as detailed in Tables 6 and 7, demonstrate that AdaMoLE consistently surpasses the performances of both the standard LoRA and MoLE configurations across most benchmarks. These experiments are designed to probe AdaMoLE's robustness and its capability to generalize effectively when applied to different underlying model architectures. The results not only affirm the superior performance of AdaMoLE in enhancing the accuracy on a variety of tasks, but also highlight its generalizability across models, showcasing its potential as a versatile tool for fine-tuning LLMs across diverse settings.

| Model | Gating | CommonsenseQA | ScienceQA | BoolQ | COPA |
|---|---|---|---|---|---|
| LoRA | - | 80.51% | 91.56% | 88.50% | **98.00%** |
| MoLE | top-2 | 80.02% | 91.28% | 89.08% | 97.00% |
| AdaMoLE | 0-1/$N$ | **81.00%** | **91.93%** | **89.94%** | **98.00%** |

Table 6: Experiment results with the Gemma-7B foundation model, where $N$ is the number of experts in one MoE module and the evaluation metric is accuracy.

| Model | Gating | CommonsenseQA | ScienceQA | BoolQ | COPA |
|---|---|---|---|---|---|
| LoRA | - | 79.77% | 89.04% | 86.33% | 93.00% |
| MoLE | top-2 | 80.67% | 90.16% | **87.46%** | 93.00% |
| AdaMoLE | 0-1/$N$ | **81.74%** | **90.67%** | 87.00% | **95.00%** |

Table 7: Experiment results with the Llama-2-13B foundation model, where $N$ is the number of experts in one MoE module and the evaluation metric is accuracy.

