# OpenReview forum: "AdaMoLE: Fine-Tuning Large Language Models with Adaptive Mixture of Low-Rank Adaptation Experts"
_colmweb.org/COLM/2024/Conference — COLM_

### Official Review · Reviewer_yPvd · 2024-04-28

**Rating:** 6
**Confidence:** 4
**Ethics Flag:** 1

**Summary:**

The paper introduces AdaMoLE, a method using an Adaptive Mixture of Low-Rank Adaptation (LoRA) Experts. Unlike traditional methods that use a static top-k strategy for activating experts, AdaMoLE incorporates a dynamic approach by utilizing a threshold network that adjusts the activation threshold based on tasks. This method employs a gating function combined with the adaptive threshold mechanism to select and activate the most suitable experts according to the input context.

**Questions To Authors:**

It would be beneficial if the authors could extend their experiments to include additional foundation models, such as Gemma.

**Reasons To Accept:**

1. The empirical results showing that AdaMoLE surpasses baseline performances on multiple benchmarks indicate substantial gains in effectiveness, primarily attributed to the adaptive selection of experts tailored to specific contextual needs.

**Reasons To Reject:**

1. The experiments are exclusively conducted using a single type of foundation model, LLama2, which may limit the generalizability of the findings. Expanding the range of models tested could provide a more comprehensive understanding of how the proposed methods perform across different foundational architectures.

2. The proposed method's approach to selecting a hyperparameter using a network-provided threshold has previously been explored. Furthermore, given that using a threshold for expert selection is already established in the literature, the introduction of a network to determine this threshold does not substantially advance the existing methodologies.

---

> ### Author Rebuttal · Authors · 2024-05-31
>
> Thank you for your thoughtful and constructive feedback. We appreciate the time and effort you have invested in reviewing our paper.
>
> **Additional Foundation Models**
>
> We conducted additional experiments using Gemma-7B and LLama-2-13B to evaluate the performance of AdaMoLE across different foundational architectures:
>
> Gemma-7B:
>
> | Model | Gating | CommonsenseQA | ScienceQA | BoolQ | COPA |
> |---|---|---|---|---|---|
> | LoRA | - | 80.51% | 91.56% | 88.50% | **98.00%** |
> | MoLE | top-2 | 80.02% | 91.28% | 89.08% | 97.00% |
> | AdaMoLE | 0-1/N | **81.00%** | **91.93%** | **89.94%** | **98.00%** |
>
> LLama-2-13B:
>
> | Model | Gating | CommonsenseQA | ScienceQA | BoolQ | COPA |
> |---|---|---|---|---|---|
> | LoRA | - | 79.77% | 89.04% | 86.33% | 93.00% |
> | MoLE | top-2 | 80.67% | 90.16% | **87.46%** | 93.00% |
> | AdaMoLE | 0-1/N | **81.74%** | **90.67%** | 87.00% | **95.00%** |
>
> These results indicate that AdaMoLE consistently outperforms both LoRA and MoLE, supporting the robustness and generalizability of our proposed method.
>
> **Threshold Mechanism**
>
> We appreciate your observation regarding the threshold mechanism. While the concept of using a threshold for expert selection is established in the literature, our contribution lies in the adaptive nature of the threshold, determined dynamically by a threshold network. This approach contrasts with static or manually tuned thresholds in previous works, offering a more flexible and context-sensitive mechanism that improves performance across varying tasks. Our empirical results demonstrate notable performance gains, validating the effectiveness of our method.
>
> | Model | Gating | CommonsenseQA | COPA |
> |---|---|---|---|
> | MoLE | top-2 | 77.15% | 92.00% |
> | AdaMoLE | 0-1/2N | 78.95% | 96.00% |
> | AdaMoLE | 0-1/N | 78.71% | 94.00% |
> | AdaMoLE | 0-3/2N | 77.15% | 92.00% |
> | AdaMoLE | 0-2/N | 74.45% | 89.00% |
>
> Averaged numbers of experts activated:
>
> | Gating | CommonsenseQA | COPA |
> |---|---|---|
> | 0-1/2N | 6.59 | 6.86 |
> | 0-1/N | 3.46 | 4.56 |
> | 0-3/2N | 1.26 | 1.71 |
> | 0-2/N | 0.33 | 0.33 |
>
> The above results show that the 1/N threshold uses more experts than top-2 but yields better results, while 3/2N uses fewer experts than MoLE top-2 but achieves comparable performance.
>
> We hope these additional experiments address your concerns and strengthen the comprehensiveness of our paper. Thank you again for your valuable feedback and the opportunity to improve our work.

---

> > ### Comment · Reviewer_yPvd · 2024-06-05
> >
> > The authors' responses addressed my concerns related to foundational models. Hence, I increased my score.

---

> > > ### Author Response · Authors · 2024-06-05
> > >
> > > Thank you for your positive feedback and for acknowledging our responses to your concerns. We appreciate your time and consideration in reviewing our work.

---

### Official Review · Reviewer_eBvp · 2024-05-11

**Rating:** 6
**Confidence:** 4
**Ethics Flag:** 1

**Summary:**

This paper proposes a novel adaptive mixture of experts LoRa tuning solution. The key contribution lies in the introduction of a learnable gate function and a threshold mechanism. Specifically, the authors develop a novel threshold function, which consists of a single linear layer followed by a sigmoid function, and it is modulated by a hyperparameter scalar, denoted as $\tau_{max}$. The experiments, conducted using the LLAMA 7B model with a LoRa rank set to 32, demonstrate that this adaptive mixture of experts LoRa sometimes outperforms the established baselines.

**Reasons To Accept:**

The proposed solution is both intuitive and innovative.
The results analysis offers some insightful findings, enhancing the value of the study.
Additionally, the writing is clear and easy to follow.

**Reasons To Reject:**

The experiments presented in this paper require further improvement to enhance the soundness of the results:

Lack of Relevant Baselines: The paper does not include comparisons against key LoRa baselines and mixture of experts LoRa baselines, such as SiRA [1]. The absence of these baselines prevents a thorough quantification of the contributions made by this study.

Limited Experimental Settings: Both LoRa and the mixture of expert LoRa are sensitive to the rank and number of experts. The authors should explore different hyperparameter settings for LoRa and mixture of expert LoRa to fully understand their impacts. Additionally, the current results suggest that performance is particularly sensitive to the hyperparameter $\tau_{max}$ . It is recommended that the authors test a wider range of settings and provide more comprehensive findings and insights to strengthen the validity of their results. Lastly, it is essential to conduct experiments using different pretrained models. Testing the proposed solution across a variety of models will help validate its effectiveness and generalizability in diverse settings.

[1] SiRA: Sparse Mixture of Low Rank Adaptation

---

> ### Author Rebuttal · Authors · 2024-05-31
>
> Thank you for your detailed and insightful feedback. We appreciate the opportunity to address your comments and strengthen our paper.
>
> **Additional Baselines**
>
> We conducted additional experiments to include SiRA and a top-3 gating strategy for MoLE:
>
> | Model | Gating | CommonsenseQA | Cosmos QA | SocialIQA | ScienceQA | RTE |
> |---|---|---|---|---|---|---|
> | LoRA | - | 76.25% | 82.71% | 76.25% | 88.67% | 73.65% |
> | SiRA | top-2 | 73.55% | 82.18% | 76.77% | 86.29% | 77.26% |
> | MoLE | top-2 | 77.15% | 83.48% | 76.92% | 88.71% | 86.28% |
> | MoLE | top-3 | 77.07% | 83.48% | 77.02% | 90.02% | 84.48% |
> | MoLE | 1/N | 75.35% | **84.69%** | 76.51% | 90.62% | 86.28% |
> | AdaMoLE | 0-1/N | **78.71%** | 84.25% | **77.28%** | **91.00%** | **87.73%** |
>
> We further tuned hyperparameters to improve baseline performances. These results confirm that AdaMoLE achieves the best or second-best performance across the benchmarks.
>
> **Hyperparameter Settings**
>
> We explored different hyperparameter settings for the number of experts and LoRA rank (N × r) on CommonsenseQA:
>
> | Model | Gating | 4 × 4 | 4 × 8 | 8 × 4| 8 × 8 | 16 × 4 |
> |---|---|---|---|---|---|---|
> | LoRA | - | 75.43% | 76.25% | 76.25% | 76.74% | 76.74% |
> | MoLE | top-2 | 73.55% | 74.45% | 77.15% | 76.82% | 76.66% |
> | MoLE | 1/N | 74.77% | 74.28% | 75.35% | 68.88% | 75.18% |
> | AdaMoLE | 0-1/N | **78.38%** | **77.89%** | **78.71%** | **78.13%** | **78.13%** |
>
> The results indicate that AdaMoLE achieves the best performance.
>
> **Maximum Thresholds**
>
> We conducted a sensitivity analysis using different threshold settings for AdaMoLE:
>
> | Model | Gating | CommonsenseQA | COPA |
> |---|---|---|---|
> | AdaMoLE | 0-1/2N | 78.95% | 96.00% |
> | AdaMoLE | 0-1/N | 78.71% | 94.00% |
> | AdaMoLE | 0-3/2N | 77.15% | 92.00% |
> | AdaMoLE | 0-2/N | 74.45% | 89.00% |
> |AdaMoLE | 0-1 | 31.94% | 57.00% |
>
> **Pretrained Models**
>
> We conducted experiments using one additional pretrained model Gemma-7B:
>
> | Model | Gating | CommonsenseQA | ScienceQA | BoolQ | COPA |
> |---|---|---|---|---|---|
> | LoRA | - | 80.51% | 91.56% | 88.50% | **98.00%** |
> | MoLE | top-2 | 80.02% | 91.28% | 89.08% | 97.00% |
> | AdaMoLE | 0-1/N | **81.00%** | **91.93%** | **89.94%** | **98.00%** |
>
> These results further demonstrate the robustness and effectiveness of AdaMoLE.
>
> We hope these additional experiments address your concerns and strengthen the validity of our study. Thank you again for your helpful feedback.

---

> > ### Comment · Reviewer_eBvp · 2024-06-05
> >
> > I would like to **increase the score from 5 to 6** due to the additional experiment results presented in the rebuttal. However, these experiments are currently not extensive. I suggest the authors further improve the experiments for the next version.

---

> > > ### Author Response · Authors · 2024-06-05
> > >
> > > Thank you for your feedback and for raising our score. We appreciate your suggestion to enhance our experiments and will work on incorporating more extensive evaluations in the next version.

---

### Official Review · Reviewer_pQdU · 2024-05-11

**Rating:** 6
**Confidence:** 4
**Ethics Flag:** 1

**Summary:**

Overview:

This paper introduces a mixture-of-experts style of low-rank adaptation fine-tuning of large language models where the learned threshold function adjusts the number of activated experts LoRA weights dynamically. The synergy of PEFT techniques and MOE architectures brings out the moderate improvement of a 7b LM on commonsense reasoning and language understanding datasets. Overall speaking, this paper is clearly written, the idea is quite straight-forward and intuitive while the experiment scope is kinda limited.

**Questions To Authors:**

Q: I am worried about the rank deficiency of MOE of LoRA matrices since the rank of sum of 8 expert matrices with rank 4 will be smaller than a single LoRA expert matrix with rank 32. If the rank collapses, the adaptability power of MOE-LoRA fine-tuning will be restricted. So is there any academic materials or instructional insights covering this?

**Reasons To Accept:**

1. Novel dynamical routing of LoRA experts when dealing with each input token by means of a learned threshold function.

**Reasons To Reject:**

1. Incomplete evaluation:

Several crucial experimental settings are missing:

a) Only commonsense reasoning and basic language understanding tasks are evaluated. Coding, mathematic reasoning and advanced LU and QA tasks (eg. MMLU and GPQA) are not involved.

b) Only one moderate-sized model (llama2-7b) is tested, which brings about the doubt whether the proposed method can be well suitable for much larger language models.

c) Only one rank (ie. 32 or 4 * 8) setting is discussed.

d) \``[0, 1/(2N)]" is the best threshold interval for the proposed method. From Fig. 3, the number of actual activated experts is quite large (almost >=6 for all layers). It seems that the computational efficiency and GPU memory occupation of AdaMoLE are flawed compared to typical top-2 routing mechanism. Even the interval of \``[0, 1/N]" has the same problem in the shallow transformer layer.

---

> ### Author Rebuttal · Authors · 2024-05-31
>
> Thank you for your thoughtful feedback. We appreciate the opportunity to address your comments.
>
> **Additional Benchmarks**
>
> We conducted experiments on more benchmarks:
>
> | Model | Gating | ScienceQA | MathQA | MedQA | RTE |
> |---|---|---|---|---|---|
> | LoRA | - | 88.67% | 31.59% | 34.41% | 73.65% |
> | MoLE | top-2 | 88.71% | 29.41% | 35.98% | 86.28% |
> | MoLE | top-3 | 90.02% | **32.73%** | 34.96% | 84.48% |
> | MoLE | 1/N | 90.62% | 31.86% | 35.43% | 86.28% |
> | AdaMoLE | 0-1/N | **91.00%** | 30.32% | **36.06%** | **87.73%** |
>
> AdaMoLE achieves the best results for most benchmarks. Note that we did not employ any prompt engineering, which could further improve performance on some tasks. Due to lack of training sets, we did not include MMLU and GPQA here.
>
> **Larger Models**
>
> We extended our experiments with Llama-2-13B:
>
> | Model | Gating | CommonsenseQA | ScienceQA | BoolQ | COPA |
> |---|---|---|---|---|---|
> | LoRA | - | 79.77% | 89.04% | 86.33% | 93.00% |
> | MoLE | top-2 | 80.67% | 90.16% | **87.46%** | 93.00% |
> | AdaMoLE | 0-1/N | **81.74%** | **90.67%** | 87.00% | **95.00%** |
>
> These results demonstrate that AdaMoLE maintains its effectiveness.
>
> **Hyperparameter Settings**
>
> We explored various numbers of experts and LoRA ranks (N × r) on CommonsenseQA:
>
> | Model | Gating | 4 × 4 | 4 × 8 | 8 × 4| 8 × 8 | 16 × 4 |
> |---|---|---|---|---|---|---|
> | LoRA | - | 75.43% | 76.25% | 76.25% | 76.74% | 76.74% |
> | MoLE | top-2 | 73.55% | 74.45% | 77.15% | 76.82% | 76.66% |
> | MoLE | 1/N | 74.77% | 74.28% | 75.35% | 68.88% | 75.18% |
> | AdaMoLE | 0-1/N | **78.38%** | **77.89%** | **78.71%** | **78.13%** | **78.13%** |
>
> These results indicate that AdaMoLE achieves the best performance.
>
> **Computational Efficiency**
>
> We conducted a sensitivity analysis to address concerns about computational efficiency:
>
> | Model | Gating | CommonsenseQA | COPA |
> |---|---|---|---|
> | MoLE | top-2 | 77.15% | 92.00% |
> | AdaMoLE | 0-1/2N | 78.95% | 96.00% |
> | AdaMoLE | 0-1/N | 78.71% | 94.00% |
> | AdaMoLE | 0-3/2N | 77.15% | 92.00% |
>
> Averaged numbers of experts activated:
>
> | Gating | CommonsenseQA | COPA |
> |---|---|---|
> | 0-1/2N | 6.59 | 6.86 |
> | 0-1/N | 3.46 | 4.56 |
> | 0-3/2N | 1.26 | 1.71 |
>
> The 1/2N threshold gives the best performance, while 1/N and 3/2N thresholds balance performance and computation. This shows AdaMoLE can dynamically adjust expert allocation efficiently.
>
> The theoretical study about rank deficiency is underway due to limited space here. Thank you again for your insightful review.

---

> > ### Author Response · Authors · 2024-06-06
> >
> > We appreciate your concern regarding the rank deficiency of MoE-LoRA matrices. To clarify, let's denote $N$ as the number of experts and $r$ as the rank of each LoRA matrix $A_i$ and $B_i$. For each input $x$, the combined MoE-LoRA matrix is given by $\sum_{i=1}^N w_i(x) B_i A_i$. While each $B_i A_i$ has a rank $r$, the combined rank of their weighted sum can be up to $N \times r$ if they span independent subspaces, as $\text{rank}\left(\sum_{i=1}^N w_i(x) B_i A_i\right) \leq \sum_{i=1}^N \text{rank}(B_i A_i) = N \times r$.
> >
> > In the case of top-k gating (or threshold gating), only the top-k experts are selected for each input, resulting in a rank of at most $k \times r$. Although $k \times r$ is smaller than $N \times r$, empirical results show that MoE-LoRA models still achieve high performance. This is because the gating mechanism dynamically selects the most relevant experts for each input, where each small matrix $B_i A_i$ specializes in different features. The dynamic selection mechanism ensures that the most relevant experts are activated based on the input $x$, allowing the model to effectively use the most informative subspaces tailored to the specific input. Consequently, even with a lower theoretical rank, the adaptive combination of small LoRA matrices can still outperform a single large LoRA matrix.
> >
> > We hope this explanation clarifies our approach. We appreciate your review and look forward to any further questions or comments you may have.

---

> > ### Comment · Reviewer_pQdU · 2024-06-06
> >
> > Appreciate the authors' additional experiments w.r.t. the evaluation issues. Adaptive selection to adjust the lora experts' parameters is a synergy of already well-known approaches. I will maintain my rating since the novelty lies in the combination rather than creation of methods. However, I would like to respect the opinions of other reviewers and chairs.

---

> > ### Comment · Reviewer_pQdU · 2024-06-06
> >
> > In the above authors' hyper-parameter test, AdaMoLE seems to saturate with the increase of lora expert number or rank, while other methods tend to have a higher performance with the total theoretical rank (i.e. $N\times{r}$). However, the overall competitiveness of proposed method is quite obvious.
> >
> > Thanks for the authors' response to my rank-deficiency concern of extending lora fine-tuning effectiveness by moe architecture. However, the response is almost an empirical standpoint which may be influenced by quite a lot experimental setting issues.
> >
> > And to clarify my concern, the rank-deficiency event may at most take place 7 times when assembling 8 rank-4 lora matrices (which I would like to name as ``low-rank ensemble issue"). Maybe the rank restoring technique proposed by ReLoRA[1] can be injected in place to MoE-LoRA, e.g. refreshing lora experts' weights according to a schedule.
> >
> > [1] ReLoRA: High-Rank Training Through Low-Rank Updates: https://arxiv.org/abs/2307.05695

---

> > > ### Author Response · Authors · 2024-06-07
> > >
> > > Thank you for your detailed feedback and for acknowledging our additional experiments. We appreciate your insights on the synergy of existing methods and the constructive suggestion regarding the rank-restoring technique from ReLoRA. We will consider these perspectives for future work. Thank you again for your review and thoughtful comments.

---

### Decision · Program_Chairs · 2024-07-10

**Decision:**

Accept

**Comment:**

The paper suggests a gating mechanism and training it in order to aggregate LoRA parameters for finetuning.
The problem is important, and the solution proposed is simple, although I'm concerned with computational complexity demands it may entail as compared to simpler approaches of linear aggregation such as chain-of-LoRA (COLA) which I recommend the authors address.
The reviewers liked the paper and I recommend acceptance.